# FROM SCARCITY TO EFFICIENCY: IMPROVING CLIP TRAINING VIA VISUAL-ENRICHED CAPTIONS

## ABSTRACT

Web-crawled datasets are pivotal to the success of pre-training vision-language models, exemplified by CLIP. However, web-crawled AltTexts can be noisy and potentially irrelevant to images, thereby undermining the crucial image-text alignment. Existing methods for rewriting captions using large language models (LLMs) have shown promise on small, curated datasets like CC3M and CC12M. Nevertheless, their efficacy on massive web-captured captions is constrained by the inherent noise and randomness in such data. In this study, we address this limitation by focusing on two key aspects: data quality and data variety. Unlike recent LLM rewriting techniques, we emphasize exploiting visual concepts and their integration into the captions to improve data quality. For data variety, we propose a novel mixed training scheme that optimally leverages AltTexts alongside newly generated **V**isual-**e**nriched **C**aptions (VeC). We use CLIP as one example and adapt the method for CLIP training on large-scale web-crawled datasets, named VeCLIP. We conduct a comprehensive evaluation of VeCLIP across small, medium, and large scales of raw data. Our results show significant advantages in image-text alignment and overall model performance, underscoring the effectiveness of VeCLIP in improving CLIP training. For example, VeCLIP achieves a remarkable over 20% improvement in COCO and Flickr30k retrieval tasks under the 12M setting. For data efficiency, we also achieve a notable over 3% improvement while using only 14% of the data employed in the vanilla CLIP and 11% in ALIGN.

## 1 INTRODUCTION

Large-scale vision-language representation learning, exemplified by CLIP (Radford et al., 2021), has gained wide attention due to the transferability of knowledge learned from image-text pairs to diverse downstream tasks such as zero-shot image classification and image-text retrieval (Li et al., 2022; Jia et al., 2021; Kwon et al., 2023). CLIP training is straightforward via the image-text contrastive loss, but involves a large-scale dataset of 400 million image-text pairs crawled from the Web. Consequently, CLIP embeddings lead to consistent improvement across various downstream tasks compared to other vision pre-training methods such as SimCLR (Chen et al., 2020) and MAE (He et al., 2022). CLIP achieves success via two scalable paradigms: data and computational resources. First, the massive web-crawled data (Schuhmann et al., 2021; 2022) enable the training to be scalable and meet the requirements of data-hungry backbones (*e.g.*, ViT (Dosovitskiy et al., 2020)). Second, the simple image-text contrastive loss grants favorable scaling properties to the computational resources.

Despite the availability of vast amounts of AltTexts (the native text field from the web-crawled data), there is a noticeable "**scarcity**" of high-quality caption data. The efforts to improve the quality and the scale of pre-training datasets remain limited (Fan et al., 2023; Nguyen et al., 2023; Wu et al., 2023), especially the degree of image-caption alignment, which is the foundation of the success of vision-language model (VLM) pre-training. As shown in Figure 1, AltTexts suffer from two issues: 1) they can be noisy, uninformative, or irrelevant to the images; 2) they may not describe all visual contents in the image. For example, in the first image, we observe a house with a white roof and a porch. However, the corresponding caption only describes the address, which proves overly abstract for effective vision-language alignment in training. Our observations demonstrate that caption quality plays a pivotal role in CLIP's performance, as detailed in Table 5b and the Appendix (*e.g.*, CC3M vs. our web-crawled 3M). It is worth noting that the captions in CC3M are derived from human annotations, rendering them less amenable to scalability. To improve the data quality, recent

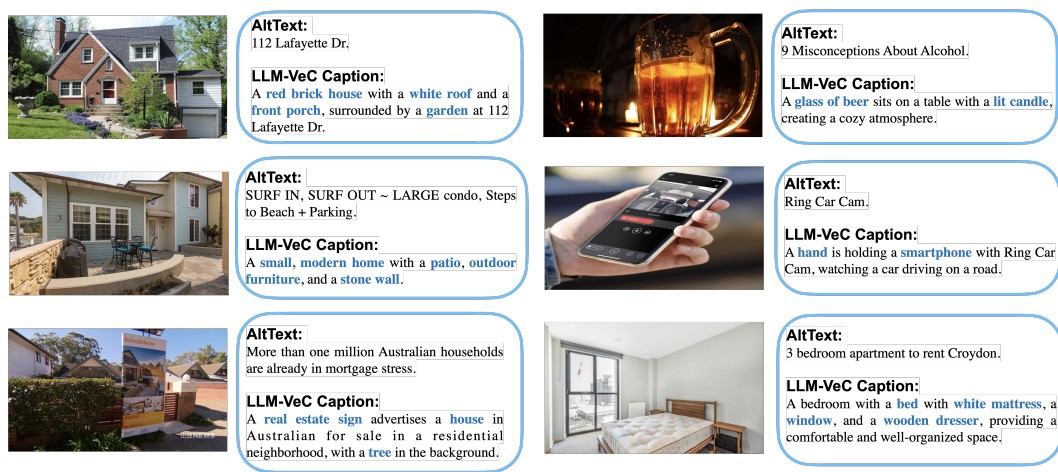

Figure 1: **Scarcity of high-quality captions in web-crawled datasets.** The AltText can be noisy and uninformative; it may not describe all visual objects present in the image. After applying our proposed LLM-VeC, new captions are imbued with more noun entities, enhancing their visual context. We retain all image-text pairs for pre-training rather than filtering out those with noisy AltTexts, as images of rich visual objects still contribute effectively to the training process.

methods propose to filter poorly aligned image-text pairs (Fan et al., 2023; Abbas et al., 2023; Cao et al., 2023; Maini et al., 2023). However, such filtering has two issues: 1) they may disregard up to 90% of data crawled from the Internet, even though a large amount of the images are suitable for training (Nguyen et al., 2023); 2) there is no universal criterion or metric existing for filtering. On the other hand, LaCLIP (Fan et al., 2023) employs Large Language Models (LLMs) to rephrase captions. However, it is restricted to CLIP training on small-scale, meticulously curated datasets like CC3M and CC12M (Changpinyo et al., 2021). Moreover, LLMs are constrained by the fact that they can only modify the syntax of sentences but cannot introduce any new image-relevant information. In other words, LaCLIP relies on the existence of high-quality captions in the pre-training datasets. High-quality datasets like manually curated CC3M and CC12M are scarce, and it is time-consuming and labor-intensive to further scale up to a larger dataset to meet the requirement of CLIP pre-training. Therefore, the scarcity of such high-quality captions poses an additional hurdle in the pursuit of efficient CLIP pre-training.

In addition to data quality, the diversity of data significantly impacts VLM pre-training (Nguyen et al., 2023). Methods relying on LLM-based rewriting may diminish data variety, given that LLMs tend to apply a uniform style in their sentence rephrasing. Moreover, existing works mainly focus on image augmentations, while texts are disregarded and unaltered during training without augmentation (Fan et al., 2023). This may also incur overfitting issues as the text encoders repeatedly encounter the same texts in each epoch. Since these techniques have exclusively undergone assessment on meticulously curated datasets like CC3M and CC12M (Changpinyo et al., 2021), their suitability for extensive, uncensored web-crawled data remains uncertain. Consequently, there is a pressing need to devise a scalable approach to enhance data quality, diversity, and training methodologies to improve pre-training for VLMs on both model performance and data "**efficiency**".

Concurrently, alongside the evolution of CLIP, there has been substantial progress in the development of instruction fine-tuned LLMs. These models and their multimodal extensions have demonstrated outstanding performance, surpassing human capabilities in various natural language and vision tasks. Inspired by these models, we investigate the potential of utilizing them to improve the noisy captions gathered from the Internet. Specifically, we initially employ LLaVA (Liu et al., 2023), a Language-Vision Assistant, to leverage visual concepts extracted from the images, denoted as **V**isual-**e**nriched **C**aptions (VeC). Given that AltTexts may lack informativeness, our objective is to integrate the newly derived visual concepts into the caption. However, it is worth noting that LLaVA (Liu et al., 2023) fine-tuned its language decoder on its own generated dataset, potentially losing its ability to accommodate comprehensive instructions. Consequently, we further propose to utilize an LLM to refine the sentence by amalgamating VeC from LLaVA and the original AltText. This process aims to maximize image-relevant information for optimal vision-language alignment.

We denote the caption generated from LLM as LLM-VeC. For data variety, we propose VeCLIP and introduce a mixed training scheme, alternating between LLM-VeC and the original AltText. This strategy ensures that the model captures all pertinent information without oversight. We generalize this scalable pipeline to curate five pre-training datasets ranging from small-scale to large-scale up to 300M. Overall, our contributions are summarized below:

- We present a visual-enriched re-captioning technique to address the scarcity of high-quality web-crawled datasets for CLIP training. This marks the initial endeavor to leverage visual concepts extracted from images and inject them into the captioning process.

- We present a scalable pipeline capable of processing data at a scale exceeding 200M. Then, we propose VeCLIP with a mixed training scheme that uses **V**isual-**e**nriched **C**aptions to improve CLIP training on model performance.

- VeCLIP achieves markedly superior performance compared to the initial CLIP training in terms of training data efficiency, *e.g.*, we use only 5% data in training but achieve competitive results in image-text retrieval tasks, furthering indicating the efficiency of our pipeline.

## 2 RELATED WORK

**Contrastive language-image pre-training.** CLIP (Radford et al., 2021) has shown its effectiveness in acquiring transferable image representations via text supervision after large-scale pre-training. Similar models such as ALIGN (Jia et al., 2021), Florence (Yuan et al., 2021), BASIC (Pham et al., 2021) and OpenCLIP (Cherti et al., 2023) have shown impressive zero-shot image classification and image-text retrieval capabilities. SLIP (Mu et al., 2022) and DeCLIP (Li et al., 2023a) incorporate self-supervised training techniques to improve performance. CoCa (Yu et al., 2022) introduces an additional decoder alongside the contrastive loss. LiT (Zhai et al., 2022) proposes to keep a pre-trained image encoder frozen and fine-tune text encoders to improve the zero-shot transferability. Nevertheless, the majority of these subsequent studies incorporate supplementary training inputs and losses, potentially exerting adverse effects on both training efficiency and memory usage.

**Improving image-text datasets.** Given the importance of the pre-training data, many works focus on improving the datasets, such as filtering less informative image-text pairs (Fan et al., 2023; Abbas et al., 2023; Cao et al., 2023; Maini et al., 2023). However, these methods may disregard a large amount of data even though some images have rich visual concepts. An alternative approach is to rewrite the caption to enhance the alignment between texts and images. For example, LaCLIP (Fan et al., 2023) employs LLMs to perform this rewriting. Nevertheless, their evaluation was conducted on meticulously curated datasets like CC3M and CC12M (Changpinyo et al., 2021), where the initial captions were already of high quality. Hence, the advantage of employing LLM rewriting on noisy web-crawled data may be marginal, given that the input for LLM consists solely of AltText. Another direction is to leverage image captioning models to generate image descriptions (Zhu et al., 2023; Nguyen et al., 2023). However, these image captioning models may have "hallucinations" during inference, where they may output some imagination that does not exist in the image (Li et al., 2023c).

## 3 IMPROVING CLIP TRAINING VIA VISUAL-ENRICHED CAPTIONS

### 3.1 PRELIMINARY

**CLIP.** The Contrastive Language-Image Pre-training (CLIP) method has shown its effectiveness in training vision models via language supervision. Specifically, a batch of $N$ image-text pairs $\{x_I, x_T\}$ is sampled from the massive training data during each training iteration. We apply data augmentations to the images before inputting them into the vision encoder. We denote $f_I$ and $f_T$ as the normalized features extracted by the vision and text encoders, respectively. We use the contrastive loss to train the model, where the paired images and texts are treated as positive pairs and the remaining as negative samples. The training loss iterating over images can be formulated as follows:

$$L_I = -\sum_{i=1}^{N} \log \frac{\exp\left(\text{sim}(f_I(\text{aug}(x_I^i)), f_T(x_T^i))/\tau\right)}{\sum_{k=1}^{N} \exp\left(\text{sim}(f_I(\text{aug}(x_I^i)), f_T(x_T^k))/\tau\right)}, \quad (1)$$

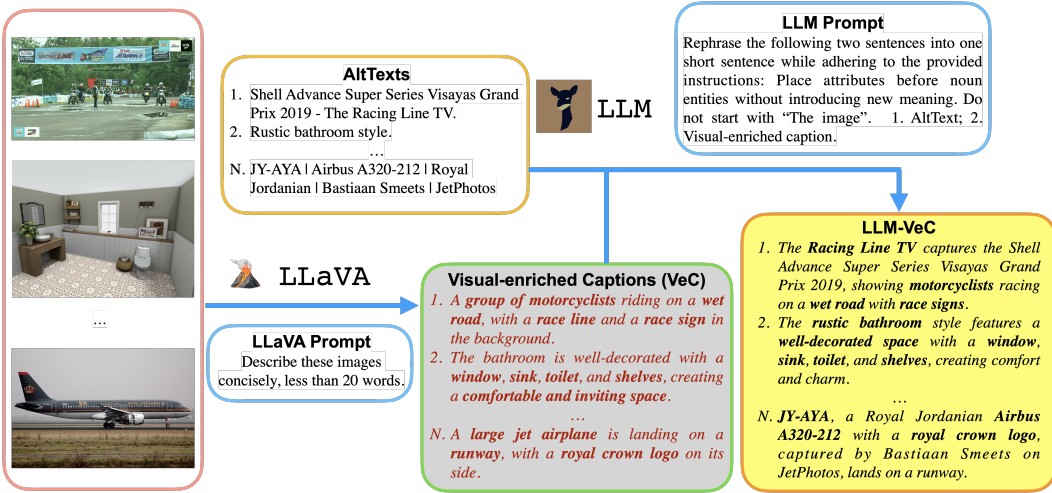

Figure 2: **An overview of the process to generate LLM-VeC.** First, we focus on exploiting visual concepts in images via leveraging a multimodal LLM (LLaVA) to describe the image with a designed prompt independent of AltText to generate Visual-enriched Captions (VeC). Second, we leverage an LLM to fuse the concepts from both AltText and VeC and subsequently generate the final caption, denoted as LLM-VeC.

where $(x_I^i, x_T^i)$ is the $i^{th}$ image-text pair in the batch, and aug$(\cdot)$ refers to image augmentations. sim$(\cdot, \cdot)$ is the similarity measurement function. We set $\tau$ as a learnable temperature parameter that scales the logits in experiments. The loss iterating over texts is symmetrical and denoted as $L_T$. Finally, the training loss is $L = (L_I + L_T)/2$.

## 3.2 Image Captioning Driven by Visual Concept Exploitation

Web-crawled captions (AltTexts) can be noisy and uninformative to the images. LaCLIP (Fan et al., 2023) used LLM to rewrite the caption, which may not be applicable if the captions are noisy as LLM can only reconstruct the sentence but cannot introduce new information without any information provided by the image. Besides LaCLIP, another branch is filtering-based methods (Fan et al., 2023; Abbas et al., 2023; Cao et al., 2023; Maini et al., 2023) targeted to remove poorly aligned image-text pairs. However, they may discard a substantial proportion of the gathered data, ranging from 60% to 90%, irrespective of the suitability of the images for training purposes. Henceforth, our objective is to build a pipeline that optimally leverages images to retain a substantial volume of data for training instead of filtering. Given the inherent noise in AltTexts, we advocate for the utilization of pre-trained multimodal models to generate augmented captions imbued with a richer array of visual concepts derived from the images. In this subsection, we use LLaVA (Liu et al., 2023) as one example.

**LLaVA and image captioning for Visual-enriched Captions (VeC).** As a multimodal model, LLaVA connects the open-set visual encoder of CLIP (Radford et al., 2021) with an LLM, such as LLaMA (Touvron et al., 2023), then fine-tune them on a visual instruction-tuning dataset. LLaVA shows its effectiveness in leveraging the capabilities of pre-trained LLM and vision foundation models. Given an input image $x_I$, we get $f_I$ from CLIP's vision encoder. Then, LLaVA applies a trainable projection matrix $\mathbf{W}$ to convert $f_I$ into language embedding tokens to achieve the image-language alignment. To mitigate the influence of AltText, we have devised AltText-independent prompts tailored for LLaVA, ensuring the full exploitation of visual concepts. We refrain from incorporating AltText information into LLaVA, while acknowledging the potential loss of pre-trained knowledge during fine-tuning of the LLM component on the generated dataset. This trade-off, however, may limit its capacity to comprehend more intricate instructions. Thus, we adopt a straightforward yet potent prompt, *"Describe the image concisely, less than 20 words"*, allowing LLaVA to generate visual concepts directly from the image autonomously. We denote this Visual-enriched Caption (VeC) generated by LLaVA as $x_{Tv}$. Subsequently, the image-text pair is converted as $(x_I, x_{Tv})$.

### 3.3 A Scalable LLM Rewriting Method for Concept Fusion

Given the constrained linguistic capacity of LLaVA, its role is confined to extracting the visual concepts from the image. Then, we employ the prowess of LLMs to refine the caption, amalgamating insights derived from both the knowledge from AltText $x_T$ and the novel visual concepts from $x_{Tv}$. This step has three main roles: 1) It ensures the retention of information delineated in AltText, thereby amplifying the informativeness of the caption; 2) It can serve as a form of "strong augmentation" in textual data, characterized by a profound restructuring of sentence syntax instead of focusing on word-level modifications used in existing language augmentation techniques (Wei & Zou, 2019; Sennrich et al., 2015); 3) It can alleviate the "hallucination" issue incurred from large vision-language models to make sure the entity described in the final caption exists in the image.

Generating rewrites for a vast corpus of texts using closed-source models like ChatGPT or Bard is impractical, considering the substantial financial costs and time incurred through API utilization. Therefore, to facilitate the rewriting tasks on a large-scale dataset, we turn to open-source state-of-the-art LLMs. Due to the license issue, we select Vicuna-1.1 (Zheng et al., 2023), renowned for its robust performance in text completion tasks, as one example of LLM rewriting in this study. We formulate a context input as the following three components. First, we include a sentence designed to apprise the LLM of the task, specifically, rewriting and fusing two attached sentences. This serves as an initial contextual cue to orient the LLM towards comprehending the overarching objective. Second, we impose several constraints on the ultimate output. For instance, our goal is to position attributes prior to noun entities, all while refraining from introducing any novel semantic interpretations. Furthermore, it is essential that the sentence refrains from commencing with the phrase "The image" and instead directly expounds upon all-encompassed concepts. Finally, the last part of the context includes two sentences ($x_v$ and $x_{Tv}$) that require fusing and rewriting, followed by the separation symbol. This ensures that the LLM is furnished with the specific texts to be fused and rewritten as part of its context input. By integrating these three components, we establish an all-encompassing context that steers the LLM towards proficiently crafting diverse and knowledge-fused text rewrites.

Employing the crafted context input as a prompt, Vicuna showcases its proficiency in executing text completion and producing rephrased renditions of the associated text samples. We conduct this process for each data point in the web-crawled image-text dataset for pre-training. Specifically, we use Vicuna-1.1-13B model to generate the final output as $x_{Tl}$. The final prompt is as follows: [*Rephrase the following two sentences into one short sentence while adhering to the provided instructions: Place attributes before noun entities without introducing new meaning. Do not start with "The image". + 1. AltText; 2. Vision-enriched caption.*] We denote the caption from LLM as LLM-VeC.

**Potential ethics of LLM and failure cases processing.** While upscaling the LLM rewriting process, we identify two scenarios in which LLM encounters difficulties in executing the designated task: 1) *Ethical Concerns.* If the AltText contains content either illegal or violent, LLM may reply, "I am sorry that I cannot..."; 2) *Length Constraint.* In cases where the AltText exceeds an optimal length, the processing time of the LLM may be significantly prolonged, thus impeding large-scale rewriting. To address the first scenario, we use VeC as the only caption to be rewritten via LLM to form LLM-VeC, thereby preemptively excluding potentially unlawful or aggressive content. In the second scenario, we mitigate this issue by preserving VeC but truncating the AltText to conform to the maximum allowable length, thus we have more visual concepts aligned with the image.

### 3.4 VeCLIP: Mixed Training Scheme with Visual-enriched Captions for CLIP

As LLM rewriting may introduce a consistent style, there could be a decline in data diversity for large-scale pre-training, even if data quality is enhanced. To enhance data diversity, we propose a mixed training scheme to serve as additional text augmentations applied in pre-training:

$$\text{mix}(x_t) \sim \text{Uniform}([x_T, x_{Tl}]). \tag{2}$$

Then, the training loss iterating over the images becomes:

$$L_I = -\sum_{i=1}^{N} \log \frac{\exp\left(\text{sim}(f_I(\text{aug}(x_I^i)), f_T(\text{mix}(x_t^i)))/\tau\right)}{\sum_{k=1}^{N} \exp\left(\text{sim}(f_I(\text{aug}(x_I^i)), f_T(\text{mix}(x_t^k)))/\tau\right)}. \tag{3}$$

The only difference with the original CLIP training is that we alternate the AltTexts with our rephrased sentences, with all other components remaining unaltered. This modification does not incur additional

Table 1: Results (Recall@$k$) on zero-shot image-to-text and text-to-image retrieval tasks on COCO and Flickr30k. 1.4B-CLIP denotes the in-house CLIP pre-trained on 1.4B web-crawled image-text pairs. We use ViT-B/16 as the vision encoder of CLIP. (*) FLIP uses ViT-L/14.

| Data | Model | COCO | | | | | | Flickr30k | | | | | |
| | | Image-to-Text | | | Text-to-Image | | | Image-to-Text | | | Text-to-Image | | |
| | | R@1 | R@5 | R@10 | R@1 | R@5 | R@10 | R@1 | R@5 | R@10 | R@1 | R@5 | R@10 |
|---|---|---|---|---|---|---|---|---|---|---|---|---|---|
| 1.8B | ALIGN (Jia et al., 2021) | 58.60 | 83.00 | 89.70 | 45.60 | 69.80 | 78.60 | 88.60 | 98.70 | 99.70 | 75.70 | 93.80 | 96.80 |
| 400M | FLIP* (Li et al., 2023b) | 60.20 | 82.60 | 89.90 | 44.20 | 69.20 | 78.40 | 89.10 | 98.50 | 99.60 | 75.40 | 92.50 | 95.90 |
| 400M | OpenAI CLIP | 53.76 | 77.92 | 85.53 | 33.09 | 58.42 | 68.90 | 88.00 | 98.70 | 99.40 | 68.70 | 90.60 | 95.20 |
| 1.4B | In-house CLIP | 61.38 | 82.80 | 89.78 | 44.48 | 69.19 | 78.28 | 87.60 | 97.90 | 98.80 | 71.70 | 91.30 | 95.24 |
| 3M | CLIP | 5.46 | 15.34 | 22.42 | 3.28 | 10.44 | 15.96 | 12.20 | 27.80 | 37.50 | 6.36 | 19.16 | 27.58 |
| | VeCLIP | 22.30 | 45.00 | 56.16 | 13.01 | 31.61 | 42.42 | 40.60 | 67.30 | 76.70 | 27.58 | 52.44 | 63.20 |
| | **Performance Gain** | +16.84 | +29.66 | +33.74 | +9.73 | +21.17 | +26.46 | +28.40 | +39.50 | +39.20 | +21.22 | +33.28 | +35.62 |
| 12M | CLIP | 24.52 | 48.28 | 59.82 | 14.28 | 34.52 | 46.29 | 44.70 | 71.80 | 80.40 | 29.06 | 58.62 | 70.00 |
| | VeCLIP | 47.78 | 72.54 | 81.56 | 31.62 | 57.19 | 68.47 | 73.90 | 92.30 | 95.90 | 55.68 | 80.78 | 87.64 |
| | **Performance Gain** | +23.26 | +24.26 | +21.74 | +17.34 | +22.67 | +22.18 | +29.20 | +20.50 | +15.50 | +26.62 | +22.16 | +17.64 |
| 100M | CLIP | 47.24 | 72.34 | 81.56 | 30.61 | 56.49 | 67.91 | 74.40 | 93.20 | 96.70 | 57.16 | 88.12 | 88.98 |
| | VeCLIP | 64.82 | 85.56 | 91.98 | 46.12 | 71.19 | 80.23 | 89.30 | 97.70 | 99.20 | 73.10 | 89.12 | 93.14 |
| | **Performance Gain** | +17.58 | +13.22 | +10.42 | +15.51 | +14.70 | +12.32 | +14.90 | +4.50 | +2.50 | +15.94 | +1.00 | +4.16 |
| | **Performance Gain vs 1.4B-CLIP** | +3.44 | +2.76 | +2.20 | +1.64 | +2.00 | +1.95 | +1.70 | -0.20 | +0.40 | +1.60 | -2.18 | -2.10 |
| 200M | CLIP | 52.20 | 76.22 | 85.04 | 34.97 | 60.42 | 71.08 | 80.90 | 94.90 | 97.60 | 63.26 | 86.58 | 92.26 |
| | VeCLIP | 67.20 | 87.28 | 92.70 | 48.40 | 73.26 | 81.79 | 91.10 | 98.50 | 99.70 | 76.32 | 93.50 | 96.40 |
| | **Performance Gain** | +15.00 | +11.06 | +7.66 | +13.43 | +12.84 | +10.71 | +10.20 | +3.60 | +2.10 | +13.06 | +6.92 | +4.14 |
| | **Performance Gain vs 1.4B-CLIP** | +5.82 | +4.48 | +2.92 | +3.92 | +4.07 | +3.51 | +3.50 | +0.60 | +0.90 | +4.62 | +2.20 | +1.16 |

computational complexity or parameter overheads compared to the standard CLIP training process. Through the strategic alternation of AltTexts and our captions, we improve both the quality and diversity of the pre-training dataset without filtering any data points. This approach empowers the model to glean insights from both AltText and visually-enhanced LLM-VeC. This simple yet effective strategy elevates the training regimen for CLIP, offering a scalable framework for optimizing other vision-language pre-training efforts utilizing extensive web-crawled data.

## 4 EXPERIMENTS

### 4.1 PRE-TRAINING DATASETS AND DOWNSTREAM TASKS

**Pre-training datasets and training setup.** We conduct pre-training experiments on four scales of Web-crawled Image-Text (WIT) datasets to show the efficiency and scalability of our method. Specifically, we set 3M as small scale, 12M as medium scale, and 100M+ as large scale. We use ViT-B/16 (Dosovitskiy et al., 2020) as the vision encoder of CLIP training. Our batch size is 8,192 for small/medium scales (3M/12M), and 32,768 for large scales (100M+). For efficiency purposes, we employ **JAX** (Bradbury et al., 2018) and train models on 64 TPUs for the 3M/12M settings, whereas we utilize 512 TPUs for the 100M/200M pre-training configurations. More details can be found in the Appendix A. To show its generalizability and effectiveness, we also evaluate it on well-curated CC3M/CC12M besides our crawled noisy WIT data, as shown in our ablation studies and Appendix C.2. We evaluate all pre-trained models on the following three tasks.

**Zero-shot image classification.** We evaluate all the models on ImageNet (Deng et al., 2009), ImageNetV2 (Recht et al., 2019), and VTAB (Zhai et al., 2019). We select 9 tasks (6 from natural sets and 3 from specialized sets) that are suitable for zero-shot classification tasks such as Flowers102 (Nilsback & Zisserman, 2008) and Caltech-101 (Fei-Fei et al., 2004) as zero-shot classification tasks. We list the details in the Appendix.

**Zero-shot image-text retrieval.** We evaluate the pre-trained models on COCO (Lin et al., 2014) and Flickr30k (Plummer et al., 2015) cross-modal retrieval tasks in zero-shot settings: Image-to-Text (denoted as I2T) and Text-to-Image (T2I) retrieval. For Flickr30k, we evaluate them on the standard 1K test set. We report the results in terms of Recall@$k$ as R@1, R@5, and R@10.

**Zero-shot image-to-image retrieval.** We select GPR1200 (Schall et al., 2022) for image-to-image retrieval. GPR1200 (Schall et al., 2022) serves as a general-purpose benchmark for content-based image retrieval, encompassing subsets drawn from six different domains. It includes 1200 categories (10 images per category). Following Schall et al. (2022), we do not split images as query and index sets for evaluation. Instead, we perform retrieval of the nearest neighbor for each image and utilize the remaining images as the index set. We report the mean Average Precision (mAP).

Table 2: Image-to-image retrieval results (mAP) on 6-domain GPR1200 (Schall et al., 2022).

| Data | Model | Domain Name | | | | | | All |
| | | Land | Faces | iNat | INST | Sketch | SOP | |
|---|---|---|---|---|---|---|---|---|
| 3M | CLIP | 57.98 | 20.76 | 17.61 | 31.14 | 18.23 | 74.29 | 36.67 |
| | VeCLIP | **66.55** | **23.51** | **20.43** | **38.63** | **24.59** | **77.65** | **41.89** |
| 12M | CLIP | 74.47 | 30.65 | 23.60 | 52.15 | 30.68 | 84.25 | 49.30 |
| | VeCLIP | **79.30** | **31.72** | **25.53** | **56.65** | **41.42** | **84.69** | **53.22** |
| 100M | CLIP | **85.64** | **51.68** | 29.66 | 68.19 | 42.45 | 90.38 | 61.33 |
| | VeCLIP | 85.59 | 42.83 | **30.72** | **71.96** | **52.59** | **90.54** | **62.37** |
| 200M | CLIP | **86.96** | **56.54** | 30.95 | 71.51 | 46.03 | 90.95 | 63.83 |
| | VeCLIP | 86.40 | 48.48 | **31.72** | **73.74** | **56.52** | **91.16** | **65.67** |

Table 3: Zero-shot classification results (Top-$k$ Accuracy) on ImageNet and ImageNetV2 (Recht et al., 2019).

| Data | Model | ImageNet | | | ImageNetV2 | | |
| | | Top-1 | Top-5 | Top-10 | Top-1 | Top-5 | Top-10 |
|---|---|---|---|---|---|---|---|
| 3M | CLIP | 5.46 | 21.05 | 28.70 | 7.09 | 18.52 | 25.83 |
| | VeCLIP | **15.98** | **34.11** | **43.23** | **13.51** | **30.03** | **38.93** |
| 12M | CLIP | 31.60 | 58.80 | 69.49 | 27.03 | 52.68 | 63.37 |
| | VeCLIP | **38.11** | **66.74** | **76.36** | **32.53** | **60.16** | **70.50** |
| 100M | CLIP | 58.64 | 85.82 | 91.79 | 50.96 | 79.77 | 86.91 |
| | VeCLIP | **60.77** | **87.77** | **93.16** | **54.17** | **82.51** | **89.24** |
| 200M | CLIP | 63.72 | 89.26 | 94.11 | 56.84 | 83.50 | 89.79 |
| | VeCLIP | **64.62** | **90.27** | **94.90** | **57.67** | **85.24** | **91.62** |

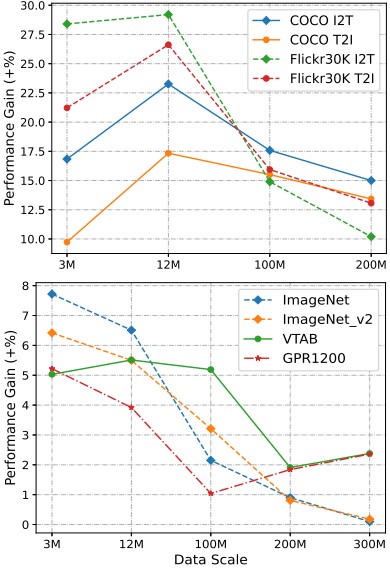

Figure 3: Performance gain on downstream tasks across different data scales.

## 4.2 RESULTS ON RETRIEVAL TASKS

**Image-to-text and text-to-image retrieval.** We summarize the main results in Table 1. We show consistent improvements across all Recall@$k$ metrics in both COCO and Flickr30k datasets for both I2T and T2I retrieval tasks. Specifically, for small and medium scales (3M/12M), we attain an improvement of +16.84%/+23.26% in Recall@1 for COCO image-to-text retrieval, respectively. Notably, the strides made in Flickr30k are particularly noteworthy, with a remarkable +28.40%/+29.20% improvement in Recall@1. Subsequently, we scale our approach to 100M and 200M, where we observe sustained and substantial improvements. Notably, we achieve a noteworthy +17.58%/+15.00% enhancement in COCO image-to-text retrieval performance using 100M and 200M, respectively. Furthermore, we observe a diminishing improvement margin as we scale up the dataset. Initially, we achieve a substantial 28.40% improvement in image-to-text retrieval for Flickr30k with the 3M dataset, which subsequently decreases to 10.20% when employing the 200M dataset. These findings show the advantages of our proposed pipeline for enhancing CLIP pre-training. By demonstrating its scalability from 3M to 200M, we provide compelling evidence of its applicability in real-world scenarios, particularly for training CLIP from scratch using WIT datasets.

**Image-to-image retrieval.** We use GPR1200 (Schall et al., 2022) with 6 domains for this setting: Google Landmarks V2 (natural and architectural landmarks) denoted as Land, IMDB Faces denoted as Faces, iNat (plants, animals, insects and fungi), INSTRE (planar images and photographs of logos/toys) denoted as INST, ImageNet Sketch denoted as Sketch, and SOP (products and objects, partly isolated). The results (mAP) are summarized in Table 2. We attain a performance gain of 5.22%/3.92% under small/medium scales (3M/12M). Even upon upscaling the dataset to 200M, we observe a notable 1.84% increase in average score across six domains. Notably, our primary performance boost is derived from the Sketch domain, underlining the crucial role of visual concepts in zero-shot transferability. Consequently, our visually-enriched captions play a pivotal role in learning such transferability towards downstream tasks.

**Data efficiency for pre-training.** In addition to benchmarking our model against a counterpart CLIP model trained at a similar scale, we include ALIGN (Jia et al., 2021), pre-trained on 1.8B data (denoted as 1.8B-ALIGN), and our in-house CLIP (Radford et al., 2021) model trained on 1.4B data (denoted as 1.4B-CLIP), as baselines operating at a significantly larger scale. We regard these models as upper bounds, each having over tenfold more data compared to our setting, to underscore the data efficiency of VeCLIP. VeCLIP shows superior performance compared to the 1.4B-CLIP model when scaling up to 100M, representing approximately 7% of its size, across nearly all downstream tasks. Specifically, in COCO, we observe a substantial +3.44%/+1.64% gain in Recall@1 for both retrieval

Table 4: Zero-shot classification accuracy. Top-1 accuracies (% ) of VTAB (Zhai et al., 2019) across 9 tasks (6 from natural and 3 from specialized sets) are reported. Full table can be found in Appendix.

| Data | Model | Natural Sets | | | | | | Specialized Sets | | | Average |
| | | Caltech101 | CIFAR100 | SVHN | DTD | OxPet | Flowers102 | EuroSAT | RESISC45 | Camelyon | |
|---|---|---|---|---|---|---|---|---|---|---|---|
| 3M | CLIP | 39.50 | 9.83 | **20.89** | 7.42 | 7.44 | 10.40 | **11.94** | 7.93 | 50.65 | 18.45 |
| | VeCLIP | **54.30** | **17.74** | 18.74 | **11.23** | **10.09** | **22.75** | 7.35 | **16.54** | **52.52** | **23.48** |
| 12M | CLIP | 70.43 | 30.06 | **30.11** | 30.69 | 34.51 | 33.67 | 8.87 | 30.05 | 53.46 | 35.76 |
| | VeCLIP | **70.58** | **45.10** | 23.61 | **30.90** | **36.22** | **43.94** | **27.46** | **38.09** | **55.54** | **41.27** |
| 100M | CLIP | 81.44 | 54.75 | 38.70 | 57.28 | **70.51** | 51.71 | 34.45 | 48.56 | 53.87 | 54.59 |
| | VeCLIP | **81.64** | **64.62** | **46.49** | **57.51** | 64.81 | **66.41** | **46.23** | **51.75** | **58.51** | **59.78** |
| 200M | CLIP | 82.30 | 61.87 | 42.83 | **64.29** | **75.60** | 58.67 | 46.73 | **55.59** | 59.30 | 60.79 |
| | VeCLIP | **83.14** | **68.14** | **44.93** | 61.95 | 72.61 | **68.51** | **47.36** | 55.10 | **62.59** | **62.70** |

tasks. Upon further scaling to 200M, the improvement becomes even more pronounced, reaching +5.82%/+3.92%. Furthermore, we achieve a notable +8.60%/+2.80% gain in COCO retrieval, as well as a +2.50%/+0.62% improvement in Flickr30k, when compared to the 1.8B ALIGN model. Remarkably, these improvements are achieved with only 11.1% of the data utilized in the pre-training process. These results show the data efficacy of VeCLIP.

## 4.3 RESULTS ON IMAGE CLASSIFICATION

**ImageNet.** We use the same prompt as CLIP ("A photo of a [classname].") for zero-shot evaluation on both ImageNet (Deng et al., 2009) and ImageNetV2 (Recht et al., 2019). The main results are summarized in Table 3. We report Top-1, Top-5, and Top-10 accuracies. In small and medium-scale settings, we observe a substantial improvement: +10.52%/+6.42% gains in Top-1 accuracy on ImageNet/ImageNetV2 under the 3M setting, and +6.51%/5.50% gains under the 12M setting. While the improvement becomes marginal upon scaling to 100M/200M, we still achieve noteworthy +2.07%/+3.21% and +0.90%/+0.83% gains on 100M and 200M across ImageNet and ImageNetV2, respectively. This shows the data efficiency of our pre-training approach.

**Visual Task Adaptation Benchmark (VTAB).** Besides ImageNet/ImageNetV2, we also select VTAB (Zhai et al., 2019) for evaluation. Table 4 summarizes zero-shot image classification results for both the original CLIP models and our models, utilizing the identical prompt set from CLIP. Our approach consistently achieves comparable or superior performance to CLIP across the majority of datasets. For instance, we observe an average accuracy improvement of over 5% under settings of 3M, 12M, and 100M. Even upon scaling up to 200M, we maintain a notable improvement of +1.91%. These results show great robustness on zero-shot classification tasks across different data distributions. We show the overall trend of the performance gain over the data scale in Figure 3.

## 4.4 ABLATION STUDY

**Importance of visual-enriched concepts.** Different from previous rewriting methods, our primary emphasis lies in infusing visual-enriched concepts harmonized with images. The ablation findings are summarized in Table 5a. We use 3M/12M as examples to show the performance gain in small/medium scales. The original AltText yields limited efficacy in retrieval tasks due to its intrinsic noise and limited visual information. VeC generated from LLaVA markedly bolsters retrieval outcomes, albeit at the expense of a dip in ImageNet classification performance. We posit that this discrepancy may stem from the protracted captions, which may not be optimal under ImageNet conditions. Introducing LLM-VeC engenders across-the-board enhancements in VeC performance. Intriguingly, the zero-shot ImageNet results still lag behind the original AltText. In essence, our LLM-VeC exerts a profound influence on retrieval prowess yet exerts a negative effect on classification tasks. We posit that this phenomenon arises from the following two reasons: 1) there can be a distributional shift in prompts from pre-training to zero-shot inference in ImageNet, particularly noteworthy given the extended length and augmented visual content of LLM-VeC; 2) the data diversity is hurt by LLM rewriting as LLM uses the same writing/paraphrasing style to fuse VeC and AltText.

**Importance of mixed training strategies.** To mitigate the aforementioned concerns, we propose a mixed training scheme to alternate between AltTexts and LLM-VeC to provide more data variety during pre-training. We summarize the ablation results of VeCLIP in Table 5b. First, we observe a slight performance improvement by randomly selecting one AltText in cases where multiple AltTexts are associated with an image. This practice augments data diversity during pre-training. Second,

| Data | Caption | Prompt Constraint | COCO (R@1) | | Flickr30k (R@1) | | ImageNet | ImageNetV2 |
|------|---------|-------------------|------|------|------|------|----------|------------|
| | | | I2T | T2I | I2T | T2I | | |
| WIT-3M | AltText | - | 5.18 | 3.40 | 10.50 | 6.88 | 8.02 | 6.88 |
| | VeC | - | 16.76 | **9.57** | 32.60 | 20.06 | 7.31 | 6.58 |
| | LLM-VeC | ✗ | 17.34 | 9.52 | 37.30 | 21.62 | 8.12 | 6.83 |
| | LLM-VeC | ✓ | **18.10** | 9.51 | **40.00** | **21.94** | **8.20** | **7.39** |
| WIT-12M | AltText | - | 22.58 | 14.23 | 44.40 | 30.90 | **31.14** | **25.91** |
| | VeC | - | 40.06 | 24.59 | 64.10 | 43.46 | 7.29 | 14.74 |
| | LLM-VeC | ✗ | 44.52 | **27.46** | 70.90 | 50.46 | 21.05 | 18.11 |
| | LLM-VeC | ✓ | **46.82** | 26.61 | **72.60** | **50.94** | 20.99 | 18.41 |

(a) Importance of visual-enriched concepts for data quality. We use the AltText with "Highest CLIP Score" (HCS) if multiple AltTexts exist on the same image in all settings.

| Data | AltText | LLM-VeC | Training Sampling | COCO (R@1) | | Flickr30k (R@1) | | ImageNet | ImageNetV2 |
|------|---------|---------|-------------------|------|------|------|------|----------|------------|
| | | | | I2T | T2I | I2T | T2I | | |
| WIT-3M | ✓ | ✗ | HCS | 5.18 | 3.40 | 10.50 | 6.88 | 8.02 | 6.88 |
| | ✓ | ✗ | random | 5.46 | 3.28 | 12.20 | 6.36 | 8.26 | 7.09 |
| | ✗ | ✓ | HCS | 18.10 | 9.51 | 40.00 | 21.94 | 8.20 | 7.39 |
| | ✓ | ✓ | HCS&mixed | 19.70 | 12.14 | 39.30 | 25.60 | 14.83 | 12.36 |
| | ✓ | ✓ | random&mixed | **22.30** | **13.01** | **40.60** | **27.58** | **15.98** | **13.51** |
| WIT-12M | ✓ | ✗ | HCS | 22.58 | 14.23 | 44.40 | 30.90 | 31.14 | 25.91 |
| | ✓ | ✗ | random | 23.32 | 14.28 | 44.70 | 29.06 | 31.60 | 27.03 |
| | ✗ | ✓ | HCS | 46.82 | 26.61 | 72.60 | 50.94 | 20.99 | 18.41 |
| | ✓ | ✓ | HCS&mixed | 46.00 | 31.10 | 72.50 | **56.82** | 37.45 | 32.41 |
| | ✓ | ✓ | random&mixed | **47.78** | **31.62** | **73.90** | 55.68 | **38.11** | **32.51** |
| CC3M | ✓ | ✗ | - | 13.88 | 9.64 | 26.30 | 18.04 | 14.59 | 12.52 |
| | ✓ | ✓ | random&mixed | **32.04** | **22.07** | **57.20** | **36.54** | **20.73** | **17.90** |

(b) Importance of the mixed training scheme for data variety. "HCS" refers to using the AltText with "Highest CLIP Score" while "random" refers to randomly selecting one if multiple AltTexts exist.

Table 5: Ablation study of VeCLIP. The highest score is bold, and the second is underlined. "mixed" is our proposed mixed training scheme to alternate among captions.

interchanging between AltText and LLM-VeC proves to be advantageous, not only in retaining substantial performance gains in retrieval tasks but also in markedly elevating zero-shot results on ImageNet. Lastly, leveraging all AltTexts and LLM-VeC within the mixed training approach in VeCLIP achieves superior results across nearly all settings.

**Larger backbone architecture.** We also investigate the performance of VeCLIP using a larger backbone architecture, ViT-L/14. The detailed results can be found in Appendix C.1. VeCLIP scaled up in backbone size can consistently outperform the original CLIP in all downstream tasks. These findings support the effectiveness of VeCLIP in improving CLIP pre-training, regardless of the specific underlying backbone architecture.

**Generalizability of LLM-VeC on well-curated datasets.** Besides our WIT datasets, we evaluate LLM-VeC on well-curated CC3M/CC12M. Table 5b shows CLIP achieves better performance when pre-trained on CC3M compared to pre-trained on WIT-3M, indicating the importance of high-quality captions for pre-training. With LLM-VeC to further improve the quality of CC3M's captions, CLIP can achieve significant improvement, *e.g.*, +18.16% on the I2T task of COCO and +6.14% on ImageNet, showing its generalizability on well-curated datasets. More results are in Appendix C.2.

## 5 DISCUSSION

**Conclusion.** We present a simple yet effective approach to improve the pre-training of VLM. We use CLIP as one example and name the resulting model VeCLIP. VeCLIP uses LLaVA, followed by another LLM, to generate visual-enriched captions for model training. VeCLIP is intentionally designed to be scalable and adaptable for handling extensive image-text datasets obtained from web crawling. We conduct a thorough evaluation of VeCLIP on a diverse range of raw datasets, spanning small, medium, and large scales. The results reveal a substantial performance boost, providing compelling evidence for the effectiveness of our strategy in enhancing large-scale VLM pre-training. VeCLIP can significantly reduce the computational cost and the size of training data for large models to reach competitive results as vanilla CLIP.

**Future work.** We employ CLIP as an illustrative instance to highlight the importance of aligning text and images within the training dataset. For future work, we plan to use the collected large-scale dataset to improve the pre-training of other types of VLMs. Further, LLM can generate outputs that encompass factual inaccuracies and hallucinations. Thus, we also plan to delve into more sophisticated filtering techniques to expunge such descriptions.

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

# Appendices

We provide additional details for datasets, experimental settings, results, and analysis in the supplementary material.

## A. DATASET DETAILS

**Pre-training datasets.** Instead of using well-curated datasets, we use image-AltText pairs sampled from a web-crawled dataset (Wu et al., 2023). We collect 300M image-text pairs from the Web and denote it as WIT-300M. Based on WIT-300M, we build four subsets to cover from small to large scales. Specifically, WIT-200M is a subset of WIT-300M. WIT-100M is a subset of WIT-200M. WIT-12M is a subset of WIT-100M. WIT-3M is a subset of WIT-12M.

Table A1: Details of 9 VTAB zero-shot classification datasets.

| Dataset | Metric | Categories | Train Size | Test Size |
|---|---|---|---|---|
| CIFAR-100 (Krizhevsky, 2009) | Accuracy | 100 | 50,000 | 10,000 |
| SVHN (Yuval, 2011) | Accuracy | 10 | 73,257 | 26,032 |
| DTD (Cimpoi et al., 2014) | Accuracy | 47 | 3,760 | 1,880 |
| Oxford Pets (Parkhi et al., 2012) | Mean per class | 37 | 3,680 | 3,669 |
| Caltech101 (Fei-Fei et al., 2004) | Mean per class | 102 | 3,060 | 6,085 |
| Flowers102 (Nilsback & Zisserman, 2008) | Mean per class | 102 | 2,040 | 6,149 |
| EuroSAT (Helber et al., 2018) | Accuracy | 10 | 10,000 | 5,000 |
| RESISC45 (Cheng et al., 2017) | Accuracy | 45 | 25,200 | 6,300 |
| Camelyon (Veeling et al., 2018) | Accuracy | 2 | 262,144 | 32,768 |

Table A2: Details of the pre-training hyper-parameters for CLIP training on our web-crawled datasets.

(a) Pre-training hyper-parameters on 3M.

| Config | Value |
|---|---|
| Batch size | $8,192$ |
| Optimizer | AdamW |
| Learning rate | $5 \times 10^{-4}$ |
| Weight decay | 0.5 |
| Adam $\beta$ | $\beta_1, \beta_2 = (0.9, 0.98)$ |
| Adam $\epsilon$ | $1 \times 10^{-8}$ |
| Total epochs | 40 |
| Warm up epochs | 1 |
| Learning rate schedule | cosine decay |

(b) Pre-training hyper-parameters on 12M.

| Config | Value |
|---|---|
| Batch size | $8,192$ |
| Optimizer | AdamW |
| Learning rate | $5 \times 10^{-4}$ |
| Weight decay | 0.5 |
| Adam $\beta$ | $\beta_1, \beta_2 = (0.9, 0.98)$ |
| Adam $\epsilon$ | $1 \times 10^{-8}$ |
| Total epochs | 35 |
| Warm up epochs | 1 |
| Learning rate schedule | cosine decay |

(c) Pre-training hyper-parameters on 100M.

| Config | Value |
|---|---|
| Batch size | $32,768$ |
| Optimizer | AdamW |
| Learning rate | $5 \times 10^{-4}$ |
| Weight decay | 0.2 |
| Adam $\beta$ | $\beta_1, \beta_2 = (0.9, 0.98)$ |
| Adam $\epsilon$ | $1 \times 10^{-6}$ |
| Total epochs | 32 |
| Warm up iterations | $2,000$ |
| Learning rate schedule | cosine decay |

(d) Pre-training hyper-parameters on 200M.

| Config | Value |
|---|---|
| Batch size | $32,768$ |
| Optimizer | AdamW |
| Learning rate | $5 \times 10^{-4}$ |
| Weight decay | 0.2 |
| Adam $\beta$ | $\beta_1, \beta_2 = (0.9, 0.98)$ |
| Adam $\epsilon$ | $1 \times 10^{-6}$ |
| Total epochs | 32 |
| Warm up iterations | $2,000$ |
| Learning rate schedule | cosine decay |

**VTAB datasets.** We choose 9 classification datasets suitable for zero-shot evaluation from VTAB (Zhai et al., 2019). Table A1 summarizes zero-shot image classification datasets. For both original CLIP models and our models, we use the identical prompt set from CLIP. Every class label is expanded using a collection of prompt templates, as defined by CLIP, including examples

like "A photo of a [classname]." The class embedding is then computed by taking the average of the embeddings of all such templates, followed by L2-normalization.

## B. IMPLEMENTATION DETAILS

**Pre-training hyper-parameters.** We summarize the pre-training hyper-parameters for CLIP training in Table A2. We pre-train models on up to 512 TPUs with JAX (Bradbury et al., 2018).

## C. MORE EXPERIMENTAL RESULTS

In this section, we present more detailed experimental results and our ablation studies (e.g., generalization of VeCLIP with a large backbone, public and well-curated datasets for pre-training).

### C.1. LARGER BACKBONE ARCHITECTURES

We also investigate the performance of VeCLIP using a larger backbone architecture, ViT-L/14. The comparison results are summarized in Table A3. First, VeCLIP shows a consistent improvement over CLIP employing ViT-L/14 across all downstream tasks. Second, VeCLIP utilizing ViT-L/14 surpasses its counterpart employing ViT-B/16, notably excelling in image classification tasks, achieving a notable improvement of over 5% on both ImageNet and ImageNetV2. This shows that VeCLIP has the potential to be scalable with larger backbone architectures and larger-scale datasets.

Table A3: Ablation studies on different backbones with VeCLIP. We use 200M as the pre-training dataset.

| Model | Backbone | COCO (R@1) I2T | COCO (R@1) T2I | Flickr30k (R@1) I2T | Flickr30k (R@1) T2I | ImageNet | ImageNetV2 |
|---|---|---|---|---|---|---|---|
| CLIP | ViT-B/16 | 52.20 | 34.97 | 80.90 | 63.23 | 63.72 | 56.84 |
| VeCLIP | ViT-B/16 | **67.20** | **48.40** | **91.10** | **76.32** | **64.62** | **57.67** |
| **Performance Gain** | | **+15.00** | **+13.43** | **+10.20** | **+13.06** | **+0.90** | **+0.81** |
| CLIP | ViT-L/14 | 53.92 | 37.86 | 84.60 | 66.78 | 68.51 | 61.13 |
| VeCLIP | ViT-L/14 | **69.92** | **51.32** | **92.60** | **79.04** | **69.85** | **63.54** |
| **Performance Gain** | | **+16.00** | **+13.46** | **+8.00** | **+12.26** | **+1.34** | **+2.41** |
| **VeCLIP ViT-L/14 vs B/16** | | **+2.72** | **+2.92** | **+1.50** | **+2.72** | **+5.23** | **+5.87** |

Table A4: Ablation studies on well-curated datasets (CC3M and CC12M (Changpinyo et al., 2021)) and the effect of data quality with ViT-B/16 as the vision backbone.

| Model | Model | COCO (R@1) I2T | COCO (R@1) T2I | Flickr30k (R@1) I2T | Flickr30k (R@1) T2I | ImageNet | ImageNetV2 |
|---|---|---|---|---|---|---|---|
| **WIT-3M** | CLIP | 5.18 | 3.40 | 10.50 | 6.88 | 8.02 | 6.88 |
| | VeCLIP | **22.30** | **13.01** | **40.60** | **27.58** | **15.98** | **13.51** |
| **Performance Gain** | | **+17.12** | **+9.61** | **+30.10** | **+20.70** | **+7.96** | **+6.63** |
| **CC3M** | CLIP | 13.88 | 9.64 | 26.30 | 18.04 | 14.59 | 12.52 |
| | VeCLIP | **32.04** | **22.07** | **57.20** | **36.54** | **20.73** | **17.90** |
| **Performance Gain** | | **+18.16** | **+12.43** | **+30.90** | **+18.50** | **+6.14** | **+5.38** |
| **WIT-12M** | CLIP | 22.58 | 14.23 | 44.40 | 30.90 | 31.14 | 25.91 |
| | VeCLIP | **47.78** | **31.62** | **73.90** | **55.68** | **38.11** | **32.51** |
| **Performance Gain** | | **+25.20** | **+17.39** | **+29.50** | **+24.78** | **+6.97** | **+6.60** |
| **CC12M** | CLIP | 37.96 | 24.40 | 59.70 | 44.90 | 39.24 | 34.41 |
| | VeCLIP | **53.23** | **36.90** | **75.20** | **62.10** | **45.32** | **40.21** |
| **Performance Gain** | | **+15.27** | **+12.50** | **+15.50** | **+17.20** | **+6.08** | **+5.80** |

### C.2. GENERALIZATION ON WELL-CURATED DATASETS: CC3M AND CC12M

Besides our crawled noisy WIT datasets, we also use a well-curated dataset, e.g., CC3M and CC12M (Changpinyo et al., 2021), to show the effectiveness and generalizability of our proposed approach on well-curated datasets. CC3M and CC12M (Changpinyo et al., 2021) were curated via several rounds of comprehensive refining and filtering to get high-quality image-caption pairs.

We show high-quality examples of CC3M and the comparison of CC3M's captions and WIT-3M's AltTexts in Appendix D. We present an experimental comparison between our crawled WIT datasets and well-curated CC3M/CC12M (Changpinyo et al., 2021) in this subsection.

**3M.** As shown in Table A4, CC3M outperforms WIT-3M when coupled with CLIP pre-training, yielding a notable increase of +10.70% on the COCO I2T task. Additionally, VeCLIP exhibits substantial improvement for both WIT-3M and CC3M. Notably, we achieve a remarkable over 30% improvement on the I2T task in Flickr30K, and an impressive over 5% boost on ImageNet and ImageNetV2.

**12M.** Similar to 3M settings, CC12M exhibits superior quality and attains better results in contrast to WIT-12M when utilized with CLIP and original AltTexts. VeCLIP demonstrates notable improvements for both WIT-12M and CC12M. For instance, VeCLIP yields a remarkable +12.27% increase in the I2T task of COCO, along with an impressive over 5% improvement on both ImageNet and ImageNetV2. These findings emphasize the effectiveness and generalizability of VeCLIP in both noisy web-crawled datasets and meticulously curated datasets, where a richer set of visual concepts is harnessed for pre-training.

### C.3. COMPLETE VISUAL DESCRIPTIONS VS SIMPLIFIED ENTITY REPRESENTATIONS

In Table 5b of the main paper, we note that sole training on LLM-VeC might detriment zero-shot performance in comparison to the original AltText. Conversely, our mixed training approach yields optimal outcomes. This intriguing finding propels us toward a more profound investigation of zero-shot classification tasks. Following established works (Radford et al., 2021; Fan et al., 2023), we employ an identical set of prompting templates, such as "a photo of a [CLS]" for ImageNet (Deng et al., 2009). It is conceivable that this direct and uncomplicated prompt may diverge significantly from LLM-VeC's pre-training, which encompasses a more extensive and intricate set of visual concepts. To address this, we reformulate LLM-VeC into a format as Simplified Entity Representation (SER). Specifically, we employ the NLTK package to extract entities from LLM-VeC and subsequently apply filtering to retain only noun entities, denoted as $(A, B, C...) \in U$. This transformation results in LLM-VeC being presented as "a photo of [U]", offering a concise representation of all extracted entities. The results are summarized in Table A5. Surprisingly, we find that even with SER-style captions, the zero-shot performance remains inferior to that achieved with the original AltText. We hypothesize that this discrepancy may arise from a lack of data diversity. When all sentences adhere to the same distribution, there exists a risk of overfitting in the pre-trained model, resulting in suboptimal performance in downstream tasks.

Table A5: Ablation studies on LLM-VeC and Simplied Entities Representation (SER). We use ViT-B/16 as the backbone and use 200M as the pre-trained dataset.

| Model | Caption | COCO (R@1) | | Flickr30k (R@1) | | ImageNet | ImageNetV2 |
| | | I2T | T2I | I2T | T2I | | |
|---|---|---|---|---|---|---|---|
| CLIP | AltText | 52.20 | 34.97 | 80.90 | 63.23 | 63.72 | 56.84 |
| VeCLIP | SER | 65.88 | **49.04** | 89.20 | 75.96 | 58.58 | 52.89 |
| VeCLIP | LLM-VeC | **67.20** | 48.40 | **91.10** | **76.32** | **64.62** | **57.67** |

### C.4. MAIN RESULTS WITH WIT-300M

We show the detailed results with the Web-crawled Image-Text 300M dataset (WIT-300M) here. We summarize the results on various downstream tasks in Table A6- A9. There are two major observations. First, we observe that the results obtained with a dataset size of 300M are close to those achieved with 200M for both CLIP and VeCLIP models. This suggests that a dataset scale of 200 million is sufficient for effectively training a ViT-B/16-based CLIP model. Second, VeCLIP achieves significant improvement on retrieval tasks even under 300M settings. Nevertheless, the improvement observed in ImageNet/ImageNetV2 is marginal.

Table A6: Results (Recall@$k$) on zero-shot image-to-text and text-to-image retrieval tasks on COCO and Flickr30k. 1.4B-CLIP denotes the in-house CLIP pre-trained on 1.4B web-crawled image-text pairs. We use ViT-B/16 as the vision encoder of CLIP. (*) Denote FLIP uses ViT-L/14.

| Data | Model | COCO | | | | | | Flickr30k | | | | | |
| | | Image-to-Text | | | Text-to-Image | | | Image-to-Text | | | Text-to-Image | | |
| | | R@1 | R@5 | R@10 | R@1 | R@5 | R@10 | R@1 | R@5 | R@10 | R@1 | R@5 | R@10 |
|---|---|---|---|---|---|---|---|---|---|---|---|---|---|
| 1.8B | ALIGN (Jia et al., 2021) | 58.60 | 83.00 | 89.70 | 45.60 | 69.80 | 78.60 | 88.60 | **98.70** | **99.70** | 75.70 | **93.80** | **96.80** |
| 400M | FLIP* (Li et al., 2023b) | 60.20 | 82.60 | 89.90 | 44.20 | 69.20 | 78.40 | 89.10 | 98.50 | 99.60 | 75.40 | 92.50 | 95.90 |
| 400M | OpenAI CLIP | 53.76 | 77.92 | 85.53 | 33.09 | 58.42 | 68.90 | 88.00 | **98.70** | 99.40 | 68.70 | 90.60 | 95.20 |
| 1.4B | In-house CLIP | 61.38 | 82.80 | 89.78 | 44.48 | 69.19 | 78.28 | 87.60 | 97.90 | 98.80 | 71.70 | 91.30 | 95.24 |
| **3M** | CLIP | 5.46 | 15.34 | 22.42 | 3.28 | 10.44 | 15.96 | 12.20 | 27.80 | 37.50 | 6.36 | 19.16 | 27.58 |
| | VeCLIP | **22.30** | **45.00** | **56.16** | **13.01** | **31.61** | **42.42** | **40.60** | **67.30** | **76.70** | **27.58** | **52.44** | **63.20** |
| | Performance Gain | +16.84 | +29.66 | +33.74 | +9.73 | +21.17 | +26.46 | +28.40 | +39.50 | +39.20 | +21.22 | +33.28 | +35.62 |
| **12M** | CLIP | 24.52 | 48.28 | 59.82 | 14.28 | 34.52 | 46.29 | 44.70 | 71.80 | 80.40 | 29.06 | 58.62 | 70.00 |
| | VeCLIP | **47.78** | **72.54** | **81.56** | **31.62** | **57.19** | **68.47** | **73.90** | **92.30** | **95.90** | **55.68** | **80.78** | **87.64** |
| | Performance Gain | +23.26 | +24.26 | +21.74 | +17.34 | +22.67 | +22.18 | +29.20 | +20.50 | +15.50 | +26.62 | +22.16 | +17.64 |
| **100M** | CLIP | 47.24 | 72.34 | 81.56 | 30.61 | 56.49 | 67.91 | 74.40 | 93.20 | 96.70 | 57.16 | 88.12 | 88.98 |
| | VeCLIP | **64.82** | **85.56** | **91.98** | **46.12** | **71.19** | **80.23** | **89.30** | **97.70** | **99.20** | **73.10** | **89.12** | **93.14** |
| | Performance Gain | +17.58 | +13.22 | +10.42 | +15.51 | +14.70 | +12.32 | +14.90 | +4.50 | +2.50 | +15.94 | +1.00 | +4.16 |
| **200M** | CLIP | 52.20 | 76.22 | 85.04 | 34.97 | 60.42 | 71.08 | 80.90 | 94.90 | 97.60 | 63.26 | 86.58 | 92.26 |
| | VeCLIP | **67.20** | **87.28** | **92.70** | **48.40** | **73.26** | **81.79** | **91.10** | **98.50** | **99.70** | **76.32** | **93.50** | **96.40** |
| | Performance Gain | +15.00 | +11.06 | +7.66 | +13.43 | +12.84 | +10.71 | +10.20 | +3.60 | +2.10 | +13.06 | +6.92 | +4.14 |
| **300M** | CLIP | 54.24 | 78.14 | 86.48 | 36.98 | 62.32 | 72.70 | 81.30 | 95.80 | 97.80 | 65.80 | 88.28 | 93.16 |
| | VeCLIP | **67.80** | **87.94** | **92.84** | **48.91** | **73.54** | **82.11** | **91.20** | **99.10** | **99.80** | **76.30** | **93.00** | **96.44** |
| | Performance Gain | +13.56 | +9.80 | +6.36 | +11.93 | +11.22 | +9.41 | +9.90 | +3.30 | +2.00 | +10.50 | +4.72 | +3.28 |

Table A7: Zero-shot classification accuracy. Top-1 Accuracies (%) of VTAB (Zhai et al., 2019) across 9 tasks (6 from natural and 3 from specialized sets) are reported.

| Data | Model | Natural Sets | | | | | | Specialized Sets | | | Average |
| | | Caltech101 | CIFAR100 | SVHN | DTD | OxPet | Flowers102 | EuroSAT | RESISC45 | Camelyon | |
|---|---|---|---|---|---|---|---|---|---|---|---|
| | | *Model Architecture: ViT-B/16* | | | | | | | | | |
| **3M** | CLIP | 39.50 | 9.83 | **20.89** | 7.42 | 7.44 | 10.40 | **11.94** | 7.93 | 50.65 | 18.45 |
| | VeCLIP | **54.30** | **17.74** | 18.74 | **11.23** | **10.09** | **22.75** | 7.35 | **16.54** | **52.52** | **23.48** |
| | Performance Gain | +14.80 | +7.91 | -2.15 | +3.81 | +2.65 | +12.35 | -4.59 | +8.61 | +1.87 | +5.03 |
| **12M** | CLIP | 70.43 | 30.06 | **30.11** | 30.69 | 34.51 | 33.67 | 8.87 | 30.05 | 53.46 | 35.76 |
| | VeCLIP | **70.58** | **45.10** | 23.61 | **30.90** | **36.22** | **43.94** | **27.46** | **38.09** | **55.54** | **41.27** |
| | Performance Gain | +0.15 | +15.04 | -6.50 | +0.21 | +1.71 | +10.27 | +18.59 | +8.04 | +2.08 | +5.51 |
| **100M** | CLIP | 81.44 | 54.75 | 38.70 | 57.28 | **70.51** | 51.71 | 34.45 | 48.56 | 53.87 | 54.59 |
| | VeCLIP | **81.64** | **64.62** | **46.49** | **57.51** | 64.81 | **66.41** | **46.23** | **51.75** | **58.51** | **59.78** |
| | Performance Gain | +0.20 | +9.87 | +7.79 | +0.23 | -5.70 | +14.70 | +11.78 | +3.19 | +4.64 | +5.19 |
| **200M** | CLIP | 82.30 | 61.87 | 42.83 | **64.29** | **75.60** | 58.67 | 46.73 | **55.59** | 59.30 | 60.79 |
| | VeCLIP | **83.14** | **68.14** | **44.93** | 61.95 | 72.61 | **68.51** | **47.36** | 55.10 | **62.59** | **62.70** |
| | Performance Gain | +0.84 | +6.27 | +2.10 | -2.34 | -2.99 | +9.84 | +0.63 | -0.49 | +3.29 | +1.91 |
| **300M** | CLIP | **83.58** | 63.36 | 50.04 | **66.16** | 74.30 | 61.81 | 39.95 | 56.44 | **53.94** | 61.06 |
| | VeCLIP | 83.07 | **68.37** | **50.07** | 65.98 | **75.36** | **69.71** | **48.28** | **58.09** | 51.94 | **63.43** |
| | Performance Gain | -0.51 | +5.01 | +0.03 | -0.18 | 1.06 | +7.90 | +8.33 | +1.65 | -2.00 | +2.37 |

## C.5. PERFORMANCE TREND ACROSS SCALES

Besides the performance gain, we also visualize the performance trend across data scales in pre-training. As shown in Figure A1, the performance of CLIP utilizing original AltTexts exhibits a marked surge with the increased data size: while its starting point is poor at 3M, it demonstrates swift progression up to 12M and 100M. However, once scaled beyond 100 million, the performance trend exhibits a gradual and eventually saturated growth. On the other hand, commencing with a higher baseline, VeCLIP employing LLM-VeC demonstrates substantial improvement in comparison to CLIP within small to medium scales (3M and 12M). As we progress beyond 300M, the performance gains of VeCLIP become relatively incremental but still noticeable in retrieval tasks. Both CLIP and VeCLIP reach a saturation point when scaled up to 100M: once over 100M, the performance gain becomes gradual and marginal.

## D. CAPTION QUALITY COMPARISON BETWEEN WELL-CURATED DATASETS AND WIT DATASETS

In Appendix C.2, we find CLIP performs notably better when pre-trained on CC3M compared to the case of being pre-trained on noisy crawled WIT datasets due to several rounds of filtering and refining involved in the curation of CC3M and CC12M. In this section, we show detailed captions

Table A8: Image-to-Image retrieval results (mAP) on 6-domain GPR1200.

| Data | Model | Land | Faces | iNat | INST | Sketch | SOP | All |
|------|-------|------|-------|------|------|--------|-----|-----|
| 3M | CLIP | 57.98 | 20.76 | 17.61 | 31.14 | 18.23 | 74.29 | 36.67 |
| | VeCLIP | 66.55 | 23.51 | 20.43 | 38.63 | 24.59 | 77.65 | 41.89 |
| 12M | CLIP | 74.47 | 30.65 | 23.60 | 52.15 | 30.68 | 84.25 | 49.30 |
| | VeCLIP | 79.30 | 31.72 | 25.53 | 56.65 | 41.42 | 84.69 | 53.22 |
| 100M | CLIP | 85.64 | 51.68 | 29.66 | 68.19 | 42.45 | 90.38 | 61.33 |
| | VeCLIP | 85.59 | 42.83 | 30.72 | 71.96 | 52.59 | 90.54 | 62.37 |
| 200M | CLIP | 86.96 | 56.54 | 30.95 | 71.51 | 46.03 | 90.95 | 63.83 |
| | VeCLIP | 86.40 | 48.48 | 31.72 | 73.74 | 56.52 | 91.16 | 65.67 |
| 300M | CLIP | 87.17 | 57.09 | 31.83 | 72.80 | 47.03 | 91.30 | 64.54 |
| | VeCLIP | 86.22 | 48.51 | 32.05 | 75.29 | 56.18 | 91.25 | 66.91 |

Table A9: Zero-shot classification results (Top-$k$ Accuracy) on ImageNet and ImageNetV2.

| Data | Model | ImageNet | | | ImageNetV2 | | |
|------|-------|-------|-------|--------|-------|-------|--------|
| | | Top-1 | Top-5 | Top-10 | Top-1 | Top-5 | Top-10 |
| 3M | CLIP | 5.46 | 21.05 | 28.70 | 7.09 | 18.52 | 25.83 |
| | VeCLIP | 15.98 | 34.11 | 43.23 | 13.51 | 30.03 | 38.93 |
| 12M | CLIP | 31.60 | 58.80 | 69.49 | 27.03 | 52.68 | 63.37 |
| | VeCLIP | 38.11 | 66.74 | 76.36 | 32.53 | 60.16 | 70.50 |
| 100M | CLIP | 58.64 | 85.82 | 91.79 | 50.96 | 79.77 | 86.91 |
| | VeCLIP | 60.77 | 87.77 | 93.16 | 54.17 | 82.51 | 89.24 |
| 200M | CLIP | 63.72 | 89.26 | 94.11 | 56.84 | 83.50 | 89.79 |
| | VeCLIP | 64.62 | 90.27 | 94.90 | 57.67 | 85.24 | 91.62 |
| 300M | CLIP | 65.70 | 90.55 | 94.87 | 58.58 | 85.32 | 91.35 |
| | VeCLIP | 65.71 | 91.15 | 95.36 | 58.76 | 86.31 | 91.95 |

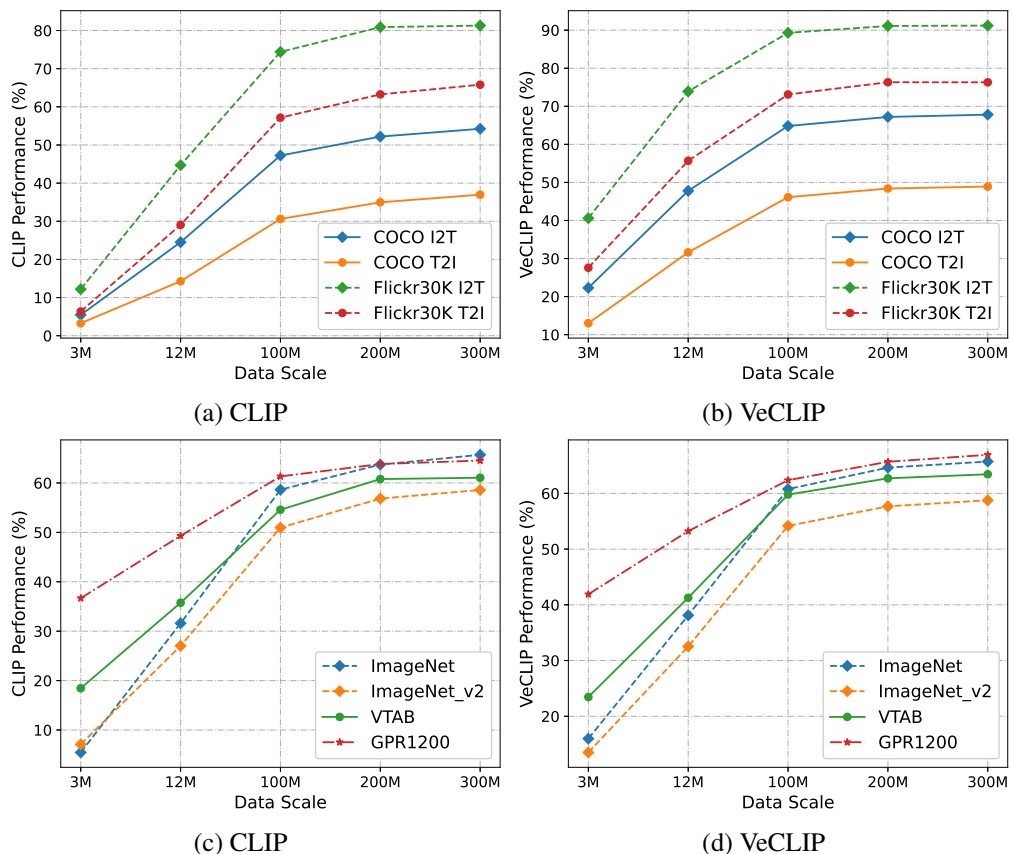

(a) CLIP

(b) VeCLIP

(c) CLIP

(d) VeCLIP

Figure A1: Performance trend with ViT-B/16 as the vision backbone. (a) and (c) show the trend of CLIP with original AltTexts while (b) and (d) show the trend of VeCLIP with LLM-VeC. The performance is improved significantly when we scale pre-training data up to 100M. Once over 100M, the performance gain becomes gradual and incremental.

from CC3M and compare them with AltTexts from WIT datasets.

**Here we provide more examples of AltText and LLM-VeC from WIT-3M:**

1. **AltText:** Ring Capri Pomellato | Pomellato Online Boutique
   **LLM-VeC:** Pomellato's Ring Capri features a delicate and elegant white stone or possibly three pearls, set against a white background.
2. **AltText:** Fiamma F45 L 450 Royal Blue Awning.
   **LLM-VeC:** The Fiamma F45 L 450 Royal Blue Awning is featured on a white car with a visible red logo for perfect closing, parked in a driveway under a tree, with a house in the background.

3. **AltText:** Union votes for strike on pensions
   **LLM-VeC:** The man with white hair, dressed in a suit and tie, exhibits a surprised or expressive look on his face, with his mouth open and hand near his face, creating a dynamic and energetic expression.
4. **AltText:** r/reallifedoodles - I can show you the world
   **LLM-VeC:** The large orange and black drone hovers in the air, carrying two small teddy bears attached to it, above a patio area, as seen in the image.
5. **AltText:** 20 Amazon Skincare Products That Keep Selling Out
   **LLM-VeC:** 20 Amazon skincare products that keep selling out feature a happy woman with dark skin, wearing a white shirt and covering her face with her hands, with a white spot or patch on her skin.
6. **AltText:** Durable White Arcane Dining Console Table With 6 Hidden Chairs
   **LLM-VeC:** A durable white arcane dining console table with 6 hidden chairs is visually appealing and ready for use, as seen in the image featuring a dining set with a white table and two benches, surrounded by black chairs.
7. **AltText:** Peaceful apartment with wi fi internet access, near old Quebec.
   **LLM-VeC:** Experience a peaceful stay in a cozy apartment with Wi-Fi internet access, located near historic Old Quebec, featuring a charming dining room with a set table and chairs on a hardwood floor, complete with a white refrigerator in the background.
8. **AltText:** CABLE BUJIA CHEVROLET CORSA 1.0 1.4 EFI FERRAZZI CABLE BUJIA CHEVROLET CORSA 1.0 1.4 EFI FERRAZZI
   **LLM-VeC:** An array of cords and wires, comprising a black rubber cable, is displayed on a pristine surface, featuring diverse configurations and orientations, with some lying horizontally and others positioned at angles.

**Here we provide more examples of original caption and LLM-VeC from CC3M:**

1. **CC3M Caption:** person runs with the ball during their training session on friday.
   **LLM-VeC:** A group of soccer players, clad in red and black jerseys, are energetically engaging in a game on a vast field, with some running and others immersed in the action, dispersed across the terrain.
2. **CC3M Caption:** a house with red roof with some bushes and a lamp post in front.
   **LLM-VeC:** A prominent two-story beige building with a distinctive tile roof stands out in the area, illuminated by a nearby lamp post. The building appears to be a complex with several houses or apartments, adding a touch of complexity to the surroundings.
3. **CC3M Caption:** eating a big sweet cupcake with chocolate at cafe.
   **LLM-VeC:** A person holds a half-eaten blueberry muffin on a plate, standing next to a dining table with a cup, while eating a big sweet cupcake with chocolate at a cafe.
4. **CC3M Caption:** paper heart with red ribbon and a bow.
   **LLM-VeC:** A pink background showcases a heart-shaped box with a bow, adorned in white with the message "Happy Valentine's Day," positioned centrally within the image.
5. **CC3M Caption:** person andactor at the premiere
   **LLM-VeC:** Two individuals, a man and a woman, are depicted standing together, both attired in formal attire. The man is donning a tuxedo with a black bow tie, while the woman is wearing a long dress. They seem to be positioning themselves for a photograph, possibly at a formal event.
6. **CC3M Caption:** wedding ceremony on the beach
   **LLM-VeC:** A picturesque wedding ceremony unfolds on a stunning white sandy beach, where perfectly arranged chairs accommodate guests in formal attire. The groom and bride exude joy and love, basking in the warm sunlight.
7. **CC3M Caption:** revenge is a dish best served cold ... with lots of lettuce .
   **LLM-VeC:** A large, possibly turtle, tortoise with an angry expression sits on rocks, displaying a saying or text message that reads "Revenge is a dish best cold served with lots of lettuce."
8. **CC3M Caption:** interior of an abandoned factory
   **LLM-VeC:** The sunlit interior of an industrial building stands in contrast to its darker exterior, with numerous windows allowing natural light to flood the space, giving it an empty and open appearance devoid of people or personal touches.

Examining the aforementioned instances, it becomes evident that CC3M's captions exhibit a notable level of precision and high quality, displaying a closer alignment with the corresponding images. Conversely, WIT-3M's AltTexts tend to be more cluttered, signaling a comparatively subpar perfor-

mance in contrast to CC3M. Upon implementing LLM-VeC, even though CC3M's captions are of high quality, they are enhanced with more visual concepts leveraged via LLM-VeC. Such integration of enriched visual concepts accounts for the significant improvement we achieve in retrieval tasks (the results are shown in Table A4).

## E. MORE EXAMPLES OF WIT WITH LLM-VeC

We conduct our scalable pipeline over 200 million image-text pairs. We randomly select more examples below to show the advantages of LLM-VeC against the original AltText in terms of visual concepts. The examples are visualized in Figure A2.

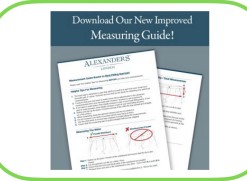

**AltText:**
Measuring Guide - Alexanders Of London.

**LLM-VeC Caption:**
Download Alexanders Of London's printable dress measuring guide with a diagram.

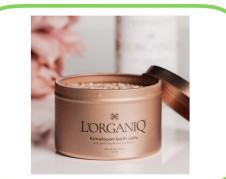

**AltText:**
Himalayan Bath Salts With Pink Clay Gift Box.

**LLM-VeC Caption:**
Himalayan Bath Salts with Pink Clay, presented in a gift box with a gold-colored metal container and a pink flower adding a touch of natural beauty.

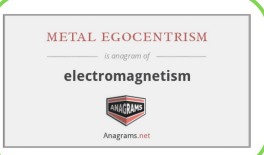

**AltText:**
metal egocentrism - electromagnetism.

**LLM-VeC Caption:**
Anagram of "electromagnetism" with a red caption at the top reading "METAL EGOCENTRISM".

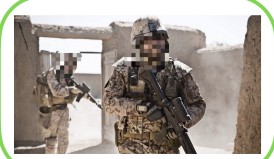

**AltText:**
Watch A War 2015 HD online.

**LLM-VeC Caption:**
Watch the 2015 HD war movie featuring two armed military men ready to protect and serve, fully equipped with weapons.

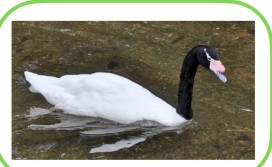

**AltText:**
Cygnus melanocoryphus / Black-necked swan in Ellen Trout Zoo.

**LLM-VeC Caption:**
A gracefully swimming black-necked swan (Cygnus melanocoryphus) creates a serene and picturesque scene in a pond, surrounded by rippling water.

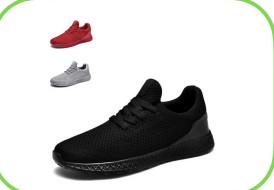

**AltText:**
Scarpe da ginnastica da corsa da uomo Casual Walking Lace Up Scarpe da ginnastica leggere • EUR 35,45.

**LLM-VeC Caption:**
Three distinct types of shoes, each boasting a unique color scheme. A black shoe with laces adorns the lineup, followed by a sleek grey shoe and a vibrant red shoe with a patterned design.

Figure A2: More examples of LLM-VeC captions and AltTexts.

