# OpenReview forum: "From Scarcity to Efficiency: Improving CLIP Training via Visual-enriched Captions"
_ICLR.cc/2024/Conference — ICLR 2024 Conference Withdrawn Submission_

### Official Review · Reviewer_kMtd · 2023-10-30

**Soundness:** 3 good
**Presentation:** 3 good
**Contribution:** 2 fair
**Rating:** 5
**Confidence:** 4

**Summary:**

The authors proposed a pipeline (LLM-VeC) for generating descriptions for web images, aiming at improving the performance of Vision & Language models. LLM-VeC uses LLaVA to generate text describing the input image, and one LLM is then used to fuse the text and AltText to yield the final corresponding text. The data mixing strategy is used when training the VL models. Extensive experiments are conducted on the well-known datasets to show the advantages of the LLM-VeC.

**Strengths:**

The main strength is that the proposed is simple and efficient. Improvements are achieved on several datasets.

**Weaknesses:**

The proposed method only combines the result from LLaVA and LLM. It is very natural and straightforward. Thus, it seems to the reviewer that there is no scientific problem solved in this paper.

**Questions:**

The proposed method is very straightforward, and it is natural to achieve improvements with more data. The targeting scientific problem is needed to be described clearly. Otherwise, the contribution might be limited.

**Details Of Ethics Concerns:**

No ethics concerns.

---

> ### Author Response · Authors · 2023-11-16
> **Response to Reviewer kMtd**
>
> We extend our gratitude for your dedicated time and effort in evaluating our submission. It's essential to clarify that our improvements stem not solely from increased data quantity, but rather from the integration of higher-quality data. Our primary aim is to introduce a cost-effective pipeline geared towards generating data beneficial for vision-language pre-training. While our approach may seem straightforward, its effectiveness should not diminish its significance in terms of technical innovation. A simpler yet impactful pipeline represents a valuable contribution, given its potential for broad application, highlighting practicality and ease of access.

---

### Official Review · Reviewer_8ucx · 2023-10-31

**Soundness:** 2 fair
**Presentation:** 3 good
**Contribution:** 2 fair
**Rating:** 5
**Confidence:** 4

**Summary:**

This paper designs a working pipeline to enrich visual caption data. Then, the generated data is used for training CLIP and significantly benefit the vision language retrieval performance. Experiments on COCO, flickr and other visual datasets show the superiority of the proposed method.

**Strengths:**

1. Data quality is important nowaday, since high-quality data always benefit the training quality. Therefore, enriching caption data is a valuable research topic.
2. The working pipeline is clear and simple to achieve the research goal.
3. Experimental results show the proposed method outperforms CLIP baseline with sufficient ablation and analysis. Different experimental settings are considered for empirical validation.

**Weaknesses:**

1. This method lacks of technical novelty. Simply using two rounds of prompt based on LLaVA and LLM to generate new caption are highly engineering operations. They actually work well based on the experiments in the draft, but I am concerning its technical contribution as a research work.
2. The experiments are mainly based on CLIP comparison, adding more backbones will further enrich this paper such as BLIP and I believe this method can easily improve performance of other backbones.

**Questions:**

Please refer to weakness for details.

---

> ### Author Response · Authors · 2023-11-16
> **Response to Reviewer 8ucx**
>
> We sincerely appreciate the time and consideration you dedicated to reviewing our submission. Our contribution centers on presenting a scalable and cost-effective pipeline aimed at augmenting CLIP training captions, facilitating data scaling to over 300M instances. It's imperative to recognize that while our approach may appear straightforward, its efficacy should not diminish its value in terms of technical novelty. A simpler yet effective pipeline constitutes a valuable contribution, as it can be readily applied across various applications, emphasizing practicality and accessibility.

---

### Official Review · Reviewer_xQRG · 2023-11-02

**Soundness:** 2 fair
**Presentation:** 3 good
**Contribution:** 2 fair
**Rating:** 3
**Confidence:** 5

**Summary:**

The research in this paper has a reasonable starting point, focusing on noise in image-text pairs, especially the issue of noise in the text part. They combine existing tuned vision-language models and LLM to rephrase caption data. In terms of the approach, there don't to be any inherent issues, and they have validated their method on various databases, including cross-modal retrieval and classification data. The results demonstrate that their data refinement approach leads to improvements in the original model. However, there are still some aspects of this paper that require further refinement and exploration.

**Strengths:**

* The idea presented in this paper is reasonable because the training corpora for models like CLIP, such as LAION, are known to be noisy. Despite some data cleaning methods, the inherent noise in the data due to web scraping and its large-scale nature remains a significant issue.
*  The approach in this article, which combines alttexts, LLM prompts, and visual-enriched captions, is also rational because this fusion of methods can effectively address the noise problem in the original data.
* Furthermore, the experimental section conducted by the authors is comprehensive, with well-justified model and test data selection, and the corresponding experimental explanations are included.

**Weaknesses:**

However, there are still some issues with this article, and these issues are directly related to the article's motivation:

- In the section on the Potential Ethics of LLM and Failure Cases Processing, the authors' viewpoint is reasonable. However, the conclusion is somewhat peculiar. In theory, LLMs like GPT-4 might only encounter issues if provided with illegal or violent inputs. It would be interesting to understand the severity of this problem when using existing models. Even when rewriting only captions, I believe the issue persists. At the very least, GPT-4 would perform sensitivity checks on inputs. I would like to know if the authors conducted a detailed analysis of this issue.

- The paper employs a strategy to choose between raw text and refined text, but I did not find a comparison with a strategy involving further pretraining or training from scratch based on the existing CLIP model. For very large models like CLIP BigG, the improvements brought by refining captions in the later stages are uncertain, especially if the model has already been trained on a substantial amount of data. It is essential to discuss the value of this approach and the improvements it can bring in the context of these larger models.

- In the ablation study section, the authors mention two explanations related to VeC. Regarding LLM's style issues, it is crucial to address how to resolve this problem since VeC's data is already biased due to the model's inherent issues. The article should propose feasible methods to tackle this problem; otherwise, the contribution of using VeC might be limited.

- Furthermore, the paper should discuss the design of prompts given AltTexts, Visual Prompts, and LLM Prompts. There are numerous possible combinations, and it is essential to explore which combination is most suitable for generating better VeC. This aspect should also be validated through human evaluation to determine the best prompt combinations.

- Although GPT-3/4 is not the primary focus of this article, at least some comparative analysis should be provided. For instance, a comparison between the new captions obtained from open-source models and those generated by GPT-3/4 could be informative.

- The paper utilizes LLaVA, which is indeed a useful method. However, there are other image attribute detection methods available, such as Recognize Anything, which can provide various visual clues. These can also be integrated into caption refinement through improved prompt design. Moreover, this approach may lead to a more focused and cleaner dataset primarily centered on visuals.

**Questions:**

Most of the issues have been discussed in the Weaknesses section above. Authors can refer to the section on Weaknesses for detailed insights.

---

> ### Author Response · Authors · 2023-11-16
> **Response to Reviewer xQRG**
>
> We are thankful for the points raised by the reviewer. Here are our responses.
>
> - Ethical issue of LLM. Harmful content often originates from either images or text, and in our case, it's the AltTexts. Prior to conducting our experiments, we first meticulously cleaned our dataset of any images containing sexual or violent content. Additionally, by integrating AltTexts with our VeC during the blending process, we further minimize the risk of harmful content infiltrating our final text. Our approach enables large language models to adhere to the VeC guidelines, especially in cases where the AltText does not align with the image or contains harmful elements. We will add the details in the main paper.
>
> - Comparison with further training on the existing CLIP model. Our primary objective is to establish a scalable pipeline for enlarging a properly aligned image-text dataset, geared towards large-scale pre-training. Our dataset encompasses over 300 million entries, and fine-tuning a model using such an extensive dataset is akin to essentially training the model from scratch. Consequently, our proposal does not advocate for utilizing the 300 million dataset entries specifically for fine-tuning purposes.
>
> - VeC bias issue. We thank the reviewer for this great point. The VeC generated from LLaVA might inadvertently inherit biases present in LLaVA, potentially resulting in captions with a consistent style. Hence, our proposition introduces a mixed training scheme that involves sampling both AltText and LLM-VeC. This approach aims to diversify the dataset and mitigate potential biases stemming from LLaVA. This strategy bears similarities to recent research, such as OpenAI's DALL-E 3, which also emphasizes diverse training methodologies to enhance dataset variety and minimize inherent biases.
>
> - Prompt issue. Yes, we investigated different prompts before we scaled data up and have validated them through human evaluation. We will add the details of the validation in Appendix.
>
> - GPT 3/4 issue. Our primary objective revolves around presenting a scalable and cost-effective re-captioning pipeline designed to handle data scaling up to millions of entries. Considering the associated costs, utilizing models like GPT-3 or GPT-4 might not be feasible within our budget constraints. Unfortunately, due to these financial limitations, we were unable to employ these models for scaling up the data and completing the comparison. However, we plan to include examples generated from GPT-3 or GPT-4 in our Appendix to offer a comparative perspective.
>
> - Visual clues. While we acknowledge the utility of image attribute detection methods like Recognize Anything in producing visual cues, their limitation lies in their inability to effectively portray relationships between distinct visual objects. Hence, our reliance on LLaVA predominantly stems from its capacity to furnish not only visual concepts but also intricate background details. In our forthcoming research, we aim to explore alternative models for extracting a more extensive array of visual concepts. This undertaking presents a significant challenge, particularly given the size of our dataset, and warrants careful consideration and effort in future investigations.